# Cost-effectiveness of HPV vaccination in 195 countries: A meta-regression analysis

Katherine L. Rosettie[1], Jonah N. Joffe[1], Gianna W. Sparks[1], Aleksandr Aravkin[1,2], Shirley Chen[1], Kelly Compton[1], Samuel B. Ewald[1], Edwin B. Mathew[1], Danielle Michael[1], Paola Pedroza Velandia[1], Molly B. Miller-Petrie[1], Lauryn Stafford[1], Peng Zheng[3], Marcia R. Weaver[1,3,4,5], Christopher J. L. Murray[1,3,4,5]*

1 Institute for Health Metrics and Evaluation, University of Washington, Seattle, WA, United States of America, 2 Departments of Applied Mathematics, University of Washington, Seattle, WA, United States of America, 3 Department of Health Metrics Sciences, University of Washington, Seattle, WA, United States of America, 4 Department of Global Health, University of Washington, Seattle, WA, United States of America, 5 Department of Health Services, University of Washington, Seattle, WA, United States of America

* cjlm@uw.edu

**Data Availability Statement:** Cost-effectiveness of HPV vaccination in 195 countries" data files are available from the GHDx database, http://ghdx.

## Abstract

Cost-effectiveness analysis (CEA) is a well-known, but resource intensive, method for comparing the costs and health outcomes of health interventions. To build on available evidence, researchers are developing methods to transfer CEA across settings; previous methods do not use all available results nor quantify differences across settings. We conducted a meta-regression analysis of published CEAs of human papillomavirus (HPV) vaccination to quantify the effects of factors at the country, intervention, and method-level, and predict incremental cost-effectiveness ratios (ICERs) for HPV vaccination in 195 countries. We used 613 ICERs reported in 75 studies from the Tufts University's Cost-Effectiveness Analysis (CEA) Registry and the Global Health CEA Registry, and extracted an additional 1,215 one-way sensitivity analyses. A five-stage, mixed-effects meta-regression framework was used to predict country-specific ICERs. The probability that HPV vaccination is cost-saving in each country was predicted using a logistic regression model. Covariates for both models included methods and intervention characteristics, and each country's cervical cancer burden and gross domestic product per capita. ICERs are positively related to vaccine cost, and negatively related to cervical cancer burden. The mean predicted ICER for HPV vaccination is 2017 US$4,217 per DALY averted (95% uncertainty interval (UI): US$773–13,448) globally, and below US$800 per DALY averted in 64 countries. Predicted ICERs are lowest in Sub-Saharan Africa and South Asia, with a population-weighted mean ICER across 46 countries of US$706 per DALY averted (95% UI: $130–2,245), and across five countries of US$489 per DALY averted (95% UI: $90–1,557), respectively. Meta-regression analyses can be conducted on CEA, where one-way sensitivity analyses are used to quantify the effects of factors at the intervention and method-level. Building on all published results, our predictions support introducing and expanding HPV vaccination, especially in countries that are eligible for subsidized vaccines from GAVI, the Vaccine Alliance, and Pan American Health Organization.

healthdata.org/ As the license states: the data are freely available for academic use and other non-commercial use. Redistribution or commercial use is not allowed without prior permission. Thus you can use the maps you made with GADM data for figures in articles published by PLoS, Springer Nature, Elsevier, MDPI, etc.

**Funding:** CJLM OPP51229 Bill & Melinda Gates Foundation https://www.gatesfoundation.org/ The funders had no role in study design, data collection and analysis, decision to publish, or preparation of the manuscript.

**Competing interests:** The authors have declared no competing interests exist.

## Introduction

Cost-effectiveness analysis (CEA) is a well-known method for comparing the costs and health outcomes of individual products or services to a standard of care, often in the context of a clinical trial. Substantial resources are needed to conduct an analysis however, making it impractical to conduct a CEA for every intervention in every setting or health care system. To build on available evidence, researchers are developing methods to transfer CEA from one setting to another. Goeree et al. summarized the factors that researchers have proposed to assess whether or not the results of a specific CEA or other economic evaluation could be transferred [1]. Among the examples they surveyed in high-income countries, most CEA results could not be transferred. Kim et al proposed a framework and checklist for transferring results of a specific CEA to a low or middle-income country [2]. In their case study, the results from one of seven economic evaluations could be transferred. Although these approaches can guide the decision to accept or reject the results of a specific CEA, they do not use all available information nor quantify differences across settings by methods, intervention characteristics, and country setting.

We conducted a meta-regression analysis of CEAs of Human papillomavirus (HPV) vaccination to estimate the effect of factors at the methods, intervention, and country-level on the incremental cost-effectiveness ratios (ICERs). The cost-effectiveness of HPV vaccination is well-studied, with more published ICERs than any other health intervention. Despite the large number of studies, there is wide variation in the results, which makes it difficult for national decision-makers to interpret the cost-effectiveness of HPV vaccination in their setting. Another challenge in leveraging the existing CEA results for HPV vaccines is the scarcity of results in super-regions with the highest cervical cancer burden. While less than 10% of published articles on CEA of HPV vaccination report estimates for Sub-Saharan Africa, the cervical cancer burden in this region is more than twice the global average [3]. The majority of published articles report estimates for high-income settings, where vaccine coverage is high and cervical cancer burden is relatively low.

In this study, we use meta-regression methods familiar to clinical evidence synthesis, and apply them to the published literature on the cost-effectiveness of HPV vaccination. Our two objectives are to: 1) identify and quantify source of heterogeneity in published CEA, and 2) predict ICERs with uncertainty intervals for 195 countries, which reflect all available published results. The ICERs include the probability that the intervention is cost-saving, meaning it both saves money and averts DALYs relative to no vaccine.

## Methods

### Human papillomavirus (HPV) vaccines

HPV is the primary cause of cervical cancer. Cervical cancer is the fourth leading cause of cancer burden among women worldwide, resulting in over eight million disability-adjusted life years (DALYs) globally in 2017 [4]. Licensed HPV vaccines include first generation bivalent and quadrivalent vaccines and a second generation nonavalent vaccine. Since the licensure of quadrivalent Gardasil in 2006, HPV vaccines have been introduced in 110 countries [5]. Universal vaccination could prevent 70–90% of HPV-related disease [6], yet coverage remains low in many low- and middle-income countries (LMICs) [7]. As of 2014, more than one-third of females ages 10–20 years had received the HPV vaccine in high-income countries, compared to only 2.7% in LMICs [7].

The World Health Organization (WHO) Director-General made a global call to action for scaling up cervical cancer prevention efforts in 2018 [8]. Subsequently, the WHO global

strategy to eliminate cervical cancer was endorsed by the Seventy-third World Health Assembly in resolution WHA73.2 [9]. One target is 90% coverage of a full HPV vaccine sequence for girls, in addition to targets for high coverage of cervical cancer screening and treatment [10]. To achieve this high level of vaccine coverage, the strategy underscores the importance of a sufficient supply of affordable HPV vaccines, introduction of HPV vaccination in countries that have not yet adopted the vaccine, and increased quality and coverage of vaccine delivery. Gavi, the Vaccine Alliance approved a plan in 2017 to accelerate introduction of HPV vaccines into national vaccine programs. They aim to support the delivery of 25 to 35 million doses of HPV vaccines annually beginning in 2021 [11]. Given the strong potential to eliminate cervical cancer with HPV vaccination coupled with cervical cancer screening and treatment, HPV vaccines are a global health priority.

## Data sources

Our data are from the Tufts University Center for the Evaluation of Value and Risk in Health registries through 2017 (Tufts registries). The Cost-Effectiveness Analysis Registry [12] contained 7287 studies that measure cost per quality-adjusted life year (QALY) gained. The Global CEA Registry [13] contained 621 studies that measure cost per disability-adjusted-life-year (DALY) averted (**Fig 1**). Between these two registries, there were 23,479 cost-effectiveness results across a wide range of interventions. Details on how these registries were compiled, including search strategies, data collection, and article review are published elsewhere [14,15]. This study complies with the Guidelines for Accurate and Transparent Health Estimates Reporting statement [16] [**S1 Table**].

## Data extraction, standardization, and mapping

Given that the Tufts registries are not compiled with the intention of being used for meta-regression analyses, additional data extraction, standardization, and mapping was necessary [**S1 and S2 Appendices**]. Missing data were extracted from articles in the Tufts CEA registries, including the age and sex of the target population, comparator and intervention descriptions, discount rates for costs and health outcomes, study time horizon, health outcomes targeted by the intervention, and study locations. Each ICER was mapped to at least one age group, sex, location, and cause that matched the categories modeled in the Global Burden of Diseases, Injuries, and Risk Factors 2017 (GBD 2017) study [3]. The age, sex, location, and cause were used to pull the total burden targeted for each intervention in the Tufts registries from DALY estimates from GBD 2017. The location and year were used to pull the gross domestic product (GDP) per capita for each intervention from GBD 2017. Each ICER for HPV vaccination was also mapped to one or more delivery platforms that were adapted from Jamison et al [17]. Five characteristics that consistently differentiated HPV vaccination interventions across articles were: vaccine cost, vaccine coverage, vaccine type (e.g. bivalent, quadrivalent), target population with respect to sex, and whether or not a booster dose was included in the vaccination schedule. When this information was not available in the Tufts registries, these data were extracted from the articles. The study currency and currency-year were used to convert all ICERs to 2017 United States dollars (US$).

The Tufts registries include a categorical variable that describes the comparator intervention, including placebo, no intervention, standard of care, and an "other" category. We defined the null comparator as no intervention (n = 1436, 79% of final sample), standard of care (n = 374, 20%), or placebo (n = 18, less than 1%). An exception was studies where the standard of care was screening for HPV infection (n = 86, 4.7% of ratios in the final sample); we defined a variable, "screening comparator" to estimate the effect of this higher standard of care for

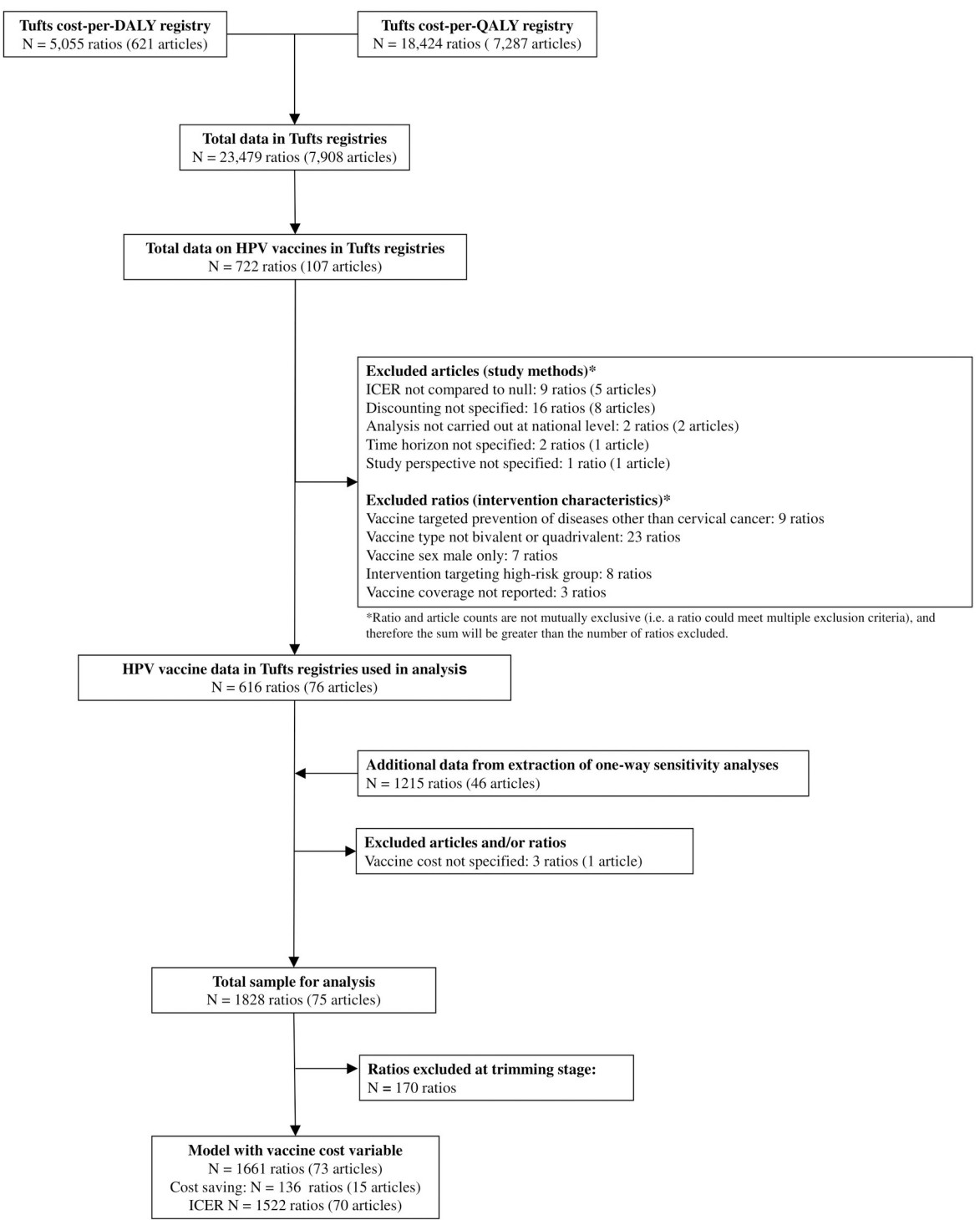

**Fig 1. Flow diagram of study selection in logistic and meta-regression models.**

cervical cancer prevention. For this study, all ICERs compared to the "other" category were re-calculated such that they were compared to the null to facilitate comparisons between ICERs. To do so, we used data in the Tufts registries on total or per person costs and total or per

person health benefits in the intervention and comparator groups. For ratios without these data reported in the registries, the necessary numerator and denominator data were extracted to re-calculate the ICERs.

## Inclusion/Exclusion criteria

Our analysis focused on HPV vaccine interventions delivered at the health center platform, as this was the most common delivery platform represented in the published estimates (98% of all HPV vaccine estimates in the Tufts registries). We excluded articles that did not clearly state the discount rates, time horizon, intervention, or comparator. We excluded ratios if they were not calculated at the country level or if the ICER could not be re-calculated relative to the null comparator. We also excluded ratios if their corresponding intervention characteristics (vaccine type, target population, coverage, cost, and booster) were either missing or represented a category for which we had insufficient data. Of the 109 ratios we excluded (6% of final sample), the two most common reasons for exclusion were vaccines that were not bivalent or quadrivalent (n = 23, 21%), and discounting not specified (n = 16, 15%).

After applying the exclusion criteria, our sample included 616 ICERs from 76 articles in 182 countries (**Fig 1**), which we refer to as main ratios [**S2 Table**]. From these 76 articles, we extracted 1,218 one-way sensitivity analyses for five covariates: vaccine cost, vaccine coverage, vaccine for females only or both sexes, cost discount rate, and discount rate for health outcomes. The one-way sensitivity analyses were matched to a reference ratio, which was often a main ratio, where the ICER of the one-way sensitivity analysis differed from the ICER of the reference ratio by the value of only one parameter. Three additional ratios from one article were excluded, because the vaccine cost was not specified.

With a total of 1,828 ICERs, 1,692 had positive incremental costs and health outcomes, and 136 were cost-saving. **Table 1** summarizes characteristics of the published studies. The majority of the 75 articles (62.7%) reported ICERs for the High income super-region, while only 8.0% and 6.7% of articles reported ICERs for Sub-Saharan Africa and South Asia, respectively. When the sensitivity analyses were included, 23.5% ICERs in the final sample were for the High income super-region, 28.1% for Sub-Saharan Africa, and 4.3% of ICERs for South Asia.

## Covariates

Three categories of covariates were used in the analysis: covariates that explain ICER variation across countries, covariates that explain ICER variation due to intervention characteristics, and covariates that explain bias in ICERs as a function of methods. Two covariates for true variation in ICERs across countries were: GDP per capita, and cervical cancer DALYs per person. GDP per capita measured the cost to the health-care system, among other things, including both the cost of the intervention and treatment costs saved when death or disability is averted. Five covariates explaining true ICER variation due to intervention characteristics were: vaccine cost measured as cost per dose multiplied by number of doses in full vaccine sequence, vaccine coverage, vaccine type (quadrivalent or bivalent), target sex (female or both males and females), vaccine coverage, and whether or not a booster dose was included in the vaccination schedule. Seven methods covariates were: perspective (health-care payer or societal/limited societal), cost discount rate, discount rate for health outcome, time horizon (lifetime or less than lifetime), outcome measure (DALYs averted or QALYs gained), comparator (screening or no intervention), and the proportion of model population with access to cervical cancer treatment (100% or less than 100%). DALYs and QALYs are not always comparable, because they are based on different methods [18,19].

**Table 1.** Descriptive statistics on sample of incremental cost-effectiveness ratios (ICERS) and articles reported in Tufts registries on human papillomavirus vaccines.

| | ICERs reported in Tufts registries or "main ratios" (%) | ICERs reported in Tufts registries plus sensitivity analyses extracted or "final sample" (%) | Articles reported in Tufts registries* (%) |
|---|---|---|---|
| **Sample size** | 613 | 1828 | 75 |
| **Study Characteristics** | | | |
| Super-region | | | |
| Central Europe, Eastern Europe, and Central Asia | 59 (9.6) | 114 (6.2) | 8 (10.7) |
| High income | 135 (22.0) | 430 (23.5) | 47 (62.7) |
| Latin America and Caribbean | 77 (12.6) | 245 (13.4) | 13 (17.3) |
| North Africa and the Middle East | 49 (8.0) | 184 (10.1) | 7 (9.3) |
| Southeast Asia, East Asia, and Oceania | 99 (16.2) | 262 (14.3) | 14 (18.7) |
| South Asia | 21 (3.3) | 79 (4.3) | 5 (6.7) |
| Sub-Saharan Africa | 173 (28.2) | 514 (28.1) | 6 (8.0) |
| Year published | | | |
| 2007 | 6 (1.1) | 22 (1.2) | 3 (4.0) |
| 2008 | 147 (23.0) | 560 (30.6) | 12 (15.8) |
| 2009 | 23 (3.6) | 69 (3.8) | 7 (9.2) |
| 2010 | 6 (0.9) | 20 (1.1) | 4 (5.3) |
| 2011 | 94 (15.4) | 173 (9.5) | 7 (9.2) |
| 2012 | 11 (1.7) | 53 (2.9) | 8 (10.5) |
| 2013 | 75 (11.9) | 538 (29.4) | 7 (9.2) |
| 2014 | 197 (31.8) | 262 (14.3) | 9 (11.8) |
| 2015 | 11 (1.7) | 73 (4.0) | 8 (10.5) |
| 2016 | 15 (2.8) | 29 (1.6) | 8 (11.8) |
| 2017 | 27 (4.6) | 29 (1.6) | 2 (2.6) |
| **Methods covariates** | | | |
| Perspective | | | |
| Societal | 11 (1.2) | 16 (0.9) | 2 (2.7) |
| Limited Societal | 138 (22.5) | 1011 (55.3) | 14 (18.7) |
| Healthcare payer | 464 (75.7) | 801 (43.8) | 59 (78.7) |
| Cost discount rate | | | |
| < 3% | 0 (0.0) | 85 (4.6) | 0 (0.0) |
| 3% | 564 (92.0) | 1508 (82.5) | 52 (69.3) |
| > 3% | 49 (8.0) | 265 (14.5) | 23 (30.7) |
| Health outcome discount rate | | | |
| < 3% | 22 (3.6) | 194 (10.6) | 9 (12.0) |
| 3% | 561 (91.5) | 1464 (80.1) | 49 (65.3) |
| > 3% | 30 (4.9) | 170 (9.3) | 17 (22.7) |
| Time Horizon | | | |
| Lifetime | 587 (95.8) | 1757 (96.2) | 65 (86.7) |
| Less than lifetime | 26 (4.2) | 71 (3.9) | 10 (13.3) |
| Health outcome measure | | | |
| QALYs | 133 (21.7) | 1319 (72.2) | 61 (81.3) |
| DALYs | 480 (78.3) | 509 (27.8) | 14 (18.7) |
| Comparator | | | |
| Null comparator | 574 (93.6) | 1742 (95.3) | 58 (77.3) |
| HPV screening | 39 (6.4) | 86 (4.7) | 17 (22.7) |

(*Continued*)

**Table 1.** (Continued)

| | ICERs reported in Tufts registries or "main ratios" (%) | ICERs reported in Tufts registries plus sensitivity analyses extracted or "final sample" (%) | Articles reported in Tufts registries* (%) |
|---|---|---|---|
| Assumption about proportion of population with access to cervical cancer treatment | | | |
| < 100% | 373 (60.8) | 857 (46.9) | 69 (92.0) |
| 100% | 240 (39.2) | 971 (53.1) | 6 (8.0) |
| **Intervention covariates** | | | |
| Type of vaccine | | | |
| Quadrivalent | 86 (15.5) | 345 (18.9) | 47 (63.2) |
| Bivalent | 527 (84.5) | 1483 (81.1) | 41 (54.0) |
| Sex | | | |
| Female only | 518 (84.5) | 1595 (87.3) | 68 (90.1) |
| Male & Female | 95 (15.5) | 233 (12.7) | 14 (18.7) |
| Booster included in vaccination schedule | | | |
| Yes | 33 (5.4) | 84 (4.6) | 17 (22.7) |
| No | 580 (94.6) | 1744 (95.4) | 71 (94.7) |
| | **Median (IQR)** | **Median (IQR)** | |
| Vaccine coverage | 70% (70, 100) | 70% (70, 80) | |
| Vaccine cost (2017 US$) | 19.9 (2.6, 223.6) | 26.5 (6.95, 180.41) | |

* denotes that the total number of articles may exceed 75, because some articles examined multiple regions, vaccine characteristics, and cost-effectiveness analyses characteristics.

ICER = Incremental cost-effectiveness ratio, DALY = disability-adjusted life year, QALY = quality-adjusted life-year.

## Modeling approaches

Our statistical analysis can be divided into two components: predicting country-specific ICERs for HPV vaccination, and predicting the probability that HPV vaccination is cost-saving in each country. We combined both sets of results to predict adjusted ICER estimates that incorporated the cost-saving probabilities.

The statistical model and fitting procedures for the analysis of ICERs was conducted in five stages, and used a mixed-effects meta-regression framework (MR-BRT) [20]. This model included priors on all covariates and a study-specific random intercept. Each stage is described briefly below; for further information, see **S3 Appendix**.

In the first stage, we estimated priors for selected covariates by leveraging the fact that one-way sensitivity analyses differ in no unmeasured covariates from their reference analyses. Four covariates had a sufficient number of sensitivity analyses reported in published CEA to estimate priors using crosswalk models: vaccine cost, vaccine coverage, cost discount rate, and discount rate for health outcomes. We matched each sensitivity analysis with its corresponding reference analysis, and the crosswalk model estimated the difference in log-ICERs between sensitivity and reference analyses as a function of the difference between values of that covariate. We then constructed Gaussian priors for these covariates to use in all subsequent stages of the analysis with means and standard deviations equal to the crosswalk parameter estimates and standard errors from these crosswalk models.

In the second stage, we estimated a nonlinear response curve for log-GDP per capita by modeling the log-ICERs as a nonlinear function of log-GDP per capita. Log cervical cancer DALYs per capita was entered linearly into this model, in addition to the four covariates addressed in the first stage, and the priors calculated in the first stage were placed on the

corresponding covariates. To make this stage more robust to model misspecification, we placed a spline ensemble on log GDP per capita. This model also used a robust statistical approach for outlier detection, and outliers trimmed at this stage were discarded from subsequent steps of the analysis. The nonlinear response curve estimated by this model was used to transform log-GDP per capita for use in subsequent stages of the analysis.

In the third stage, we selected additional covariates to include in the final meta-regression using a generalized Lasso approach for linear mixed effects models. The spline-transformed log-GDP per capita, log cervical cancer DALYs per capita, and the four crosswalk covariates, were pre-selected covariates at this stage, and the priors estimated for the crosswalk covariates were placed on those covariates. This process selected from eight additional candidate covariates: vaccine type, target sex, booster dose, perspective, time horizon, outcome measure, comparator, and the proportion of model population with access to cervical cancer treatment. Only one of these covariates, the booster dose, was not selected for inclusion in the final model.

In the fourth stage we selected the standard deviation of a Gaussian prior to apply to all regression parameters other than the intercept and the parameters for the four crosswalk covariates. To select a standard deviation, we fit a mixed effects meta-regression models with random intercepts by study, and priors on crosswalk covariates as calculated in the first stage. We normalized all other covariates and included Gaussian priors on those covariates, centered at zero and with a standard deviation that was constant across covariates. We varied this standard deviation using a grid-search and used 10-fold cross-validation to select the standard deviation that minimized the mean squared error for predicting values in the holdout set. We then converted the prior standard deviation back to the unstandardized scale for each covariate.

In the fifth stage, we fit a mixed effects model with a random intercept and priors on covariates determined in the first and fourth stages. This model included priors on covariates calculated in the first and fourth stages and the transformed version of log-GDP per capita, and random intercepts by study.

The probability that HPV vaccination was cost-saving was analyzed using a mixed effect logistic regression model. We sought to use the same covariates as the meta-regression model. The model included two country-level covariates, four intervention-level covariates, and four of seven method-level covariates: cost discount rate, discount rate for health outcomes, time horizon, and the proportion of model population with access to cervical cancer treatment. To account for between study heterogeneity, data were grouped by article, and a random intercept was calculated for each article [**S4 Appendix**].

The estimated meta-regression and logistic regression models were used to predict the ICER and probability that the vaccine was cost-saving, respectively, for each country as function of the two country-level covariates, and vaccine cost. The HPV vaccine does not have a single global market price. We used the cost of all required doses based on the 2017 cost per dose as reported to the WHO's Market Information for Access to Vaccines (MI4A) [21] and aggregated by Linksbridge [22]. We used the United Nations Childrens' Emergency Fund (UNICEF) price for the 57 Gavi-eligible countries [23], and Pan American Health Organization for 22 countries eligible for their Revolving Fund for supported countries and vaccines [**S5 Appendix**]. The three other intervention characteristics were held constant for our predictions: vaccine coverage of 70% (median across all studies), a bivalent vaccine, and target sex of females only. Our country predictions used a health sector payor perspective, 3% discount rate for costs and health outcomes, lifetime time horizon, DALYs averted as the health outcome measure, null comparator, and less than 100% access to cervical cancer treatment.

Adjusted ICER predictions were combined predictions from meta-regression and logistic regression models. The logistic regression provided the probability that HPV vaccines were cost-saving in each country. We subtracted each probability from one, and multiplied the resulting value by the mean predicted ICERs, lower bound of predicted ICERs (2.5th percentile), and upper bound of predicted ICERs (97.5th percentile) from the meta-regression analysis. This ensured that for countries with the highest probabilities of being cost-saving, ICERs were adjusted downwards (i.e. towards 0) more than for countries with lower cost-saving probabilities.

To place the results in the context of each country's economy, we also report the ICERs as one-half, one, and three times GBD per capita. ICERs can contribute to a country-specific process for interpreting results and deciding whether or not to adopt the intervention [24,25]. In the absence of this process, one times GDP per capita, and three times GDP per capita are frequently cited thresholds for categorizing interventions are very cost-effective or cost-effective, respectively, in LMIC [26]. The opportunity cost of health-care expenditures is an alternative threshold, which corresponds to one-half times GDP per capita or less in the countries where it has been estimated [27,28]. To account for uncertainty, we compare the upper bound of the 95% uncertainty interval (UI) to these thresholds.

The meta-regression analysis was performed with an open source mixed effects package https://github.com/zhengp0/limetr [20]. ICERs for 195 countries with adjustments were predicted with Python version 3.0 (Python Software Foundation, available at http://www.python.org). The logistic regression analysis was performed using the open source software lme4 in R version 4.0.5 (Comprehensive R Archive Network, available at https://cran.r-project.org/bin/windows/base/). The maps are original content that was created with open source software ggplot2 in R package 4.0.5, and map boundaries from DAGM (Database of Global Administrative Areas, available at https://gadm.org/data.html).

## Results

Beginning with the meta-regression results, we focus on three independent variables that explain true variation across ICERs: vaccine cost, burden of cervical cancer, and GDP per capita (**Fig 2**). The effects of vaccine cost, and cervical cancer burden on the ICER are in the expected direction; higher cost increases the incremental cost and leads to a higher ICER, and higher burden increases the incremental health improvements and leads to a lower ICER. The effect of GDP per capita has a slight U-shape, decreasing as health systems improve and the savings in treatment costs increase, and then increasing as the higher cost of vaccine administration exceeds the treatment saving. As explained above, four additional covariates at the intervention-level, and six at the method-level were selected for inclusion in the final model, even though the standard errors were large relative to the coefficients of some covariates (**S3 Appendix**). As expected, the ICER is negatively associated with the quadrivalent relative to the bivalent vaccine, after controlling for vaccine cost, and is positively associated with targeting the vaccine to both sexes relative to females only. The ICER increases with the discount rates. Some between-study heterogeneity is unexplained, which leads to wide UI in the estimated ICER. The ratio of upper bound divided by the lower bound of the 95% UI is between 16.8 and 17.7 in all countries (**Table 2**).

The mean probability that HPV vaccination is cost-saving from the logistic regression analysis is less than 0.1%. Congo (Brazzaville) has the highest mean probability (0.4%), and Egypt has the lowest, which is effectively zero. The predicted mean probabilities are lower than the final sample of published CEA, because the predictions use market prices of the vaccine, which

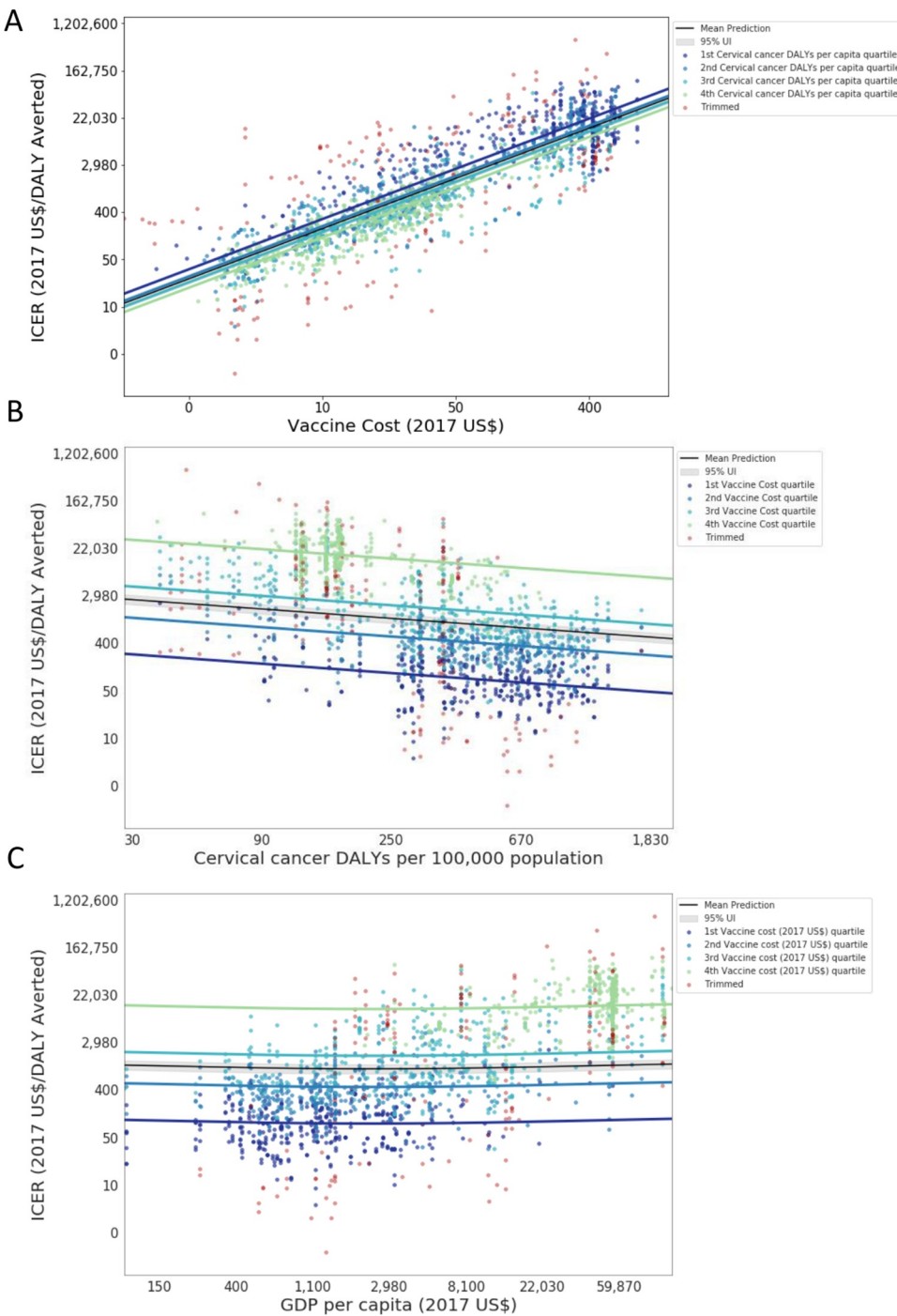

**Fig 2.** Model fit for three independent variables that explain true variation across ICERs: (A) vaccine cost, (B) cervical cancer burden, and (C) GDP per capita. Results for vaccine cost (A) are reported by GDP per capita quartile, and results for burden of cervical cancer (B) and GDP per capita (C) are reported by vaccine cost quartile, after controlling for all other method and intervention-level characteristic model covariates. The X- and Y-axes are in log-space. The grey band indicates the total uncertainty (fixed and random effects) for the mean/median burden value for GDP per capita in (A) and of vaccine cost in (B) and (C). ICER = incremental cost-effectiveness ratio; DALY = disability-adjusted-life-year; GDP = gross domestic product per capita in 2017 US$.

**Table 2. Predicted cost-effectiveness ratios by country adjusted for cost-saving probabilities.**

| Country | Predicted ICER adjusted for cost-saving probabilities (2017 US$ per DALY Averted) | Cervical cancer DALYs per 100 000 women ages 15+ years | Tufts registry dataset plus sensitivity analyses extracted | | |
|---|---|---|---|---|---|
| | | | Number of ratios | Minimum ICER (2017 US$ per DALY or QALY) | Maximum ICER (2017 US$ per DALY or QALY) |
| Central Europe Eastern Europe and Central Asia | | | | | |
| Albania | 6543 (1201 to 20,723) | 147 | 1 | 4682 | 4682 |
| Armenia | 4691 (865 to 15,020) | 342 | 9 | 33 | 1463 |
| Azerbaijan | 5557 (1021 to 17,723) | 231 | 9 | 64 | 852 |
| Belarus | 5037 (927 to 16,110) | 294 | 1 | 1476 | 1476 |
| Bosnia and Herzegovina | 4964 (914 to 15,878) | 301 | 1 | 3052 | 3052 |
| Bulgaria | 4346 (801 to 13,916) | 443 | 1 | 879 | 879 |
| Croatia | 8101 (1486 to 25,868) | 203 | 1 | 17,381 | 17,381 |
| Czech Republic | 7539 (1382 to 24,138) | 261 | 1 | 16,872 | 16,872 |
| Estonia | 7603 (1394 to 24,363) | 248 | 10 | 1964 | 16,323 |
| Georgia | 4212 (778 to 13,487) | 452 | 9 | 38 | 1327 |
| Hungary | 7292 (1339 to 23,377) | 269 | 4 | 9,971 | 50,565 |
| 'Kazakhstan | 5004 (920 to 16,053) | 326 | 1 | 653 | 653 |
| Kyrgyzstan | 464 (85 to 1482) | 326 | 9 | 26 | 1059 |
| Latvia | 7867 (1443 to 25,152) | 220 | 1 | 1111 | 1111 |
| Lithuania | 6882 (1264 to 22,038) | 314 | 1 | 853 | 853 |
| Macedonia | 5188 (955 to 16,565) | 268 | 1 | 1751 | 1751 |
| Moldova | 4712 (869 to 15,066) | 329 | 6 | 52 | 470 |
| Mongolia | 4457 (822 to 14,296) | 391 | 9 | 53 | 497 |
| Montenegro | 5286 (972 to 16,892) | 265 | 1 | 1229 | 1229 |
| Poland | 6802 (1250 to 21,808) | 317 | 1 | 10,655 | 10,655 |
| Romania | 3860 (712 to 12,254) | 618 | 1 | 684 | 684 |
| Russian Federation | 5551 (1019 to 17,766) | 254 | 1 | 931 | 931 |
| Serbia | 4209 (777 to 13,501) | 464 | 1 | 979 | 979 |
| Slovakia | 7044 (1293 to 22,529) | 303 | 1 | 10,759 | 10,759 |
| Slovenia | 9047 (1654 to 28,874) | 167 | 8 | 2105 | 34,443 |
| Tajikistan | 706 (129 to 2224) | 113 | 9 | 82 | 593 |
| Turkmenistan | 4608 (849 to 14,797) | 378 | 1 | 1874 | 1874 |
| Ukraine | 5183 (954 to 16,525) | 260 | 6 | 25 | 484 |
| Uzbekistan | 491 (90 to 1565) | 279 | 9 | 54 | 668 |
| High income | | | | | |
| Andorra | 11,056 (2013 to 35,186) | 111 | 0 | NA | NA |
| Argentina | 5607 (1036 to 17,859) | 504 | 19 | cost-saving | 14,008 |
| Australia | 11,493 (2089 to 36,622) | 108 | 1 | 28,254 | 28,254 |
| Austria | 10,244 (1867 to 32,699) | 137 | 6 | 3195 | 29,170 |
| Belgium | 10,488 (1911 to 33,440) | 128 | 5 | 5555 | 59,737 |
| Brunei | 6655 (1220 to 21,001) | 391 | 1 | 9,294 | 9294 |
| Canada | 10,150 (1849 to 32,426) | 142 | 40 | 3233 | 62,190 |
| Chile | 6750 (1240 to 21,595) | 330 | 9 | cost-saving | 21,697 |
| Cyprus | 11,040 (2013 to 35,018) | 102 | 1 | 44,310 | 44,310 |
| Denmark | 9968 (1816 to 31,808) | 152 | 8 | 2441 | 26,064 |
| Finland | 12,271 (2231 to 38,925) | 86 | 1 | 48,069 | 48,069 |
| France | 10,243 (1867 to 32,669) | 134 | 5 | 2588 | 45,703 |
| Germany | 9789 (1785 to 31,284) | 152 | 20 | cost-saving | 73,307 |

*(Continued)*

**Table 2.** (Continued)

| Country | Predicted ICER adjusted for cost-saving probabilities (2017 US$ per DALY Averted) | Cervical cancer DALYs per 100 000 women ages 15+ years | Tufts registry dataset plus sensitivity analyses extracted | | |
|---|---|---|---|---|---|
| | | | Number of ratios | Minimum ICER (2017 US$ per DALY or QALY) | Maximum ICER (2017 US$ per DALY or QALY) |
| Greece | 9200 (1682 to 29,326) | 157 | 1 | 25,733 | 25,733 |
| Greenland | 6747 (1236 to 21,231) | 395 | 0 | NA | NA |
| Iceland | 12,260 (2229 to 38,896) | 86 | 2 | 24,078 | 331,568 |
| Ireland | 10,366 (1886 to 33,028) | 143 | 27 | 3252 | 51,127 |
| Israel | 11,267 (2052 to 35,787) | 102 | 1 | 28,620 | 28,620 |
| Italy | 10,721 (1954 to 34,099) | 115 | 2 | 14,438 | 32,464 |
| Japan | 9803 (1787 to 31,338) | 154 | 28 | 76 | 66,255 |
| Luxembourg | 12,115 (2197 to 38,663) | 103 | 1 | 29,055 | 29,055 |
| Malta | 11,010 (2007 to 34,933) | 103 | 1 | 174,461 | 174,461 |
| Netherlands | 11,091 (2018 to 35,525) | 113 | 102 | 1597 | 66,507 |
| New Zealand | 11,100 (2021 to 35,297) | 108 | 38 | 537 | 137,554 |
| Norway | 11,482 (2085 to 36,6679) | 113 | 42 | cost-saving | 155,552 |
| Portugal | 8848 (1619 to 28,240) | 174 | 1 | 14,888 | 14,888 |
| Singapore | 12,196 (2216 to 38,780) | 91 | 5 | 3450 | 22,794 |
| South Korea | 10,405 (1899 to 33,070) | 119 | 1 | 36,987 | 36,987 |
| Spain | 10,434 (1903 to 33,193) | 121 | 2 | cost-saving | 26,070 |
| Sweden | 10,700 (1947 to 34,152) | 127 | 1 | 27,843 | 27,843 |
| Switzerland | 12,141 (2203 to 38,851) | 98 | 8 | 8147 | 108,426 |
| United Kingdom | 9953 (1815 to 31,773) | 144 | 17 | 4994 | 73,289 |
| Uruguay | 5718 (1054 to 18,172) | 496 | 5 | 189 | 7399 |
| USA | 27,600 (5041 to 88,339) | 153 | 29 | 1229 | 123,817 |
| Latin America and Caribbean | | | | | |
| Antigua and Barbuda | 6220 (1145 to 19,878) | 394 | 0 | NA | NA |
| Barbados | 5382 (993 to 17,045) | 583 | 5 | cost-saving | 8513 |
| Belize | 791 (146 to 2514) | 579 | 6 | 7 | 3013 |
| Bermuda | 9904 (1803 to 31,583) | 163 | 0 | NA | NA |
| Bolivia | 748 (138 to 2375) | 654 | 10 | 72 | 6116 |
| Brazil | 1010 (185 to 3212) | 350 | 44 | cost-saving | 14,618 |
| Colombia | 1028 (189 to 3290) | 316 | 21 | 21 | 77,007 |
| Costa Rica | 1084 (199 to 3465) | 286 | 5 | 50 | 5254 |
| Cuba | 928 (171 to 2962) | 401 | 10 | 41 | 5030 |
| Dominica | 752 (139 to 2376) | 685 | 0 | NA | NA |
| Dominican Republic | 923 (170 to 2941) | 412 | 5 | 121 | 4637 |
| Ecuador | 899 (165 to 2871) | 427 | 5 | 177 | 4574 |
| El Salvador | 816 (150 to 2599) | 531 | 5 | 47 | 2769 |
| Grenada | 783 (144 to 2470) | 639 | 1 | 1783 | 1783 |
| Guatemala | 857 (158 to 2738) | 465 | 5 | 198 | 5299 |
| Guyana | 725 (134 to 2295) | 721 | 10 | cost-saving | 3477 |
| Haiti | 313 (58 to 981) | 923 | 10 | 6 | 1199 |
| Honduras | 1335 (244 to 4219) | 150 | 11 | 54 | 4112 |
| Jamaica | 792 (146 to 2514) | 584 | 5 | 221 | 5539 |
| Mexico | 1026 (188 to 3270) | 330 | 23 | 11 | 11,627 |
| Nicaragua | 406 (75 to 1294) | 453 | 10 | cost-saving | 1704 |
| Panama | 6053 (1116 to 19,363) | 413 | 5 | 79 | 4139 |

*(Continued)*

**Table 2.** (*Continued*)

| Country | Predicted ICER adjusted for cost-saving probabilities (2017 US$ per DALY Averted) | Cervical cancer DALYs per 100 000 women ages 15+ years | Tufts registry dataset plus sensitivity analyses extracted | | |
|---|---|---|---|---|---|
| | | | Number of ratios | Minimum ICER (2017 US$ per DALY or QALY) | Maximum ICER (2017 US$ per DALY or QALY) |
| Paraguay | 762 (141 to 2418) | 638 | 5 | 10 | 1829 |
| Peru | 913 (168 to 2913) | 417 | 19 | 18 | 11,906 |
| Puerto Rico | 8344 (1527 to 26,712) | 207 | 0 | NA | NA |
| Saint Lucia | 798 (147 to 2522) | 603 | 1 | 1703 | 1703 |
| Saint Vincent and the Grenadines | 671 (124 to 2108) | 911 | 1 | 1810 | 1810 |
| Suriname | 760 (140 to 2403) | 667 | 5 | 134 | 4743 |
| The Bahamas | 5777 (1062 to 18,219) | 526 | 5 | 247 | 13,617 |
| Trinidad and Tobago | 5819 (1072 to 18,471) | 485 | 5 | 91 | 6684 |
| Venezuela | 820 (151 to 2600) | 553 | 8 | cost-saving | 532 |
| Virgin Islands | 6339 (1164 to 20,035) | 426 | 0 | NA | NA |
| North Africa and Middle East | | | | | |
| Afghanistan | 503 (92 to 1606) | 279 | 16 | 107 | 9160 |
| Algeria | 6369 (1169 to 20,202) | 160 | 8 | cost-saving | 5111 |
| Bahrain | 13,391 (2436 to 42,171) | 61 | 8 | cost-saving | 95,797 |
| Egypt | 10,057 (1854 to 31,337) | 48 | 8 | 317 | 26,238 |
| Iran | 9222 (1683 to 28,936) | 66 | 12 | 264 | 30,513 |
| Iraq | 10,822 (1982 to 33,711) | 43 | 8 | 253 | 21,350 |
| Jordan | 10,438 (1921 to 32,502) | 44 | 8 | 273 | 26,860 |
| Kuwait | 17,504 (3169 to 55,442) | 33 | 8 | cost-saving | 24,990 |
| Lebanon | 4196 (1246 to 15,793) | 82 | 8 | 212 | 28,506 |
| Libya | 6466 (1186 to 20,501) | 155 | 8 | 58 | 19,631 |
| Morocco | 5317 (979 to 16,942) | 245 | 8 | cost-saving | 5214 |
| Oman | 12,329 (2247 to 38,867) | 72 | 8 | cost-saving | 29,319 |
| Palestine | 9632 (1773 to 30,034) | 54 | 0 | NA | NA |
| Qatar | 16,070 (2907 to 51,447) | 47 | 8 | cost-saving | 617,462 |
| Saudi Arabia | 15,911 (2888 to 49,937) | 39 | 8 | 279 | 68,586 |
| Sudan | 801 (146 to 2509) | 81 | 12 | 142 | 6001 |
| Syria | 954 (174 to 2972) | 52 | 8 | 299 | 25,081 |
| Tunisia | 7982 (1461 to 25,110) | 89 | 8 | 37 | 8577 |
| Turkey | 9122 (1663 to 28,721) | 73 | 8 | 99 | 18,827 |
| United Arab Emirates | 11,291 (2055 to 35,898) | 105 | 8 | cost-saving | 103,848 |
| Yemen | 733 (134 to 2311) | 110 | 16 | 183 | 44,035 |
| South Asia | | | | | |
| Bangladesh | 538 (98 to 1708) | 226 | 20 | 5 | 6238 |
| Bhutan | 5877 (1080 to 18,661) | 189 | 9 | 19 | 1099 |
| India | 471 (86 to 1502) | 311 | 25 | cost-saving | 6176 |
| Nepal | 500 (92 to 1595) | 279 | 9 | 26 | 346 |
| Pakistan | 619 (113 to 1957) | 157 | 16 | cost-saving | 2142 |
| Southeast Asia East Asia and Oceania | | | | | |
| American Samoa | 5256 (966 to 16,817) | 276 | 0 | NA | NA |
| Cambodia | 422 (77 to 1345) | 418 | 9 | 55 | 1788 |
| China | 5614 (1032 to 17,908) | 228 | 19 | 296 | 154,065 |

(*Continued*)

**Table 2.** (Continued)

| Country | Predicted ICER adjusted for cost-saving probabilities (2017 US$ per DALY Averted) | Cervical cancer DALYs per 100 000 women ages 15+ years | Tufts registry dataset plus sensitivity analyses extracted | | |
|---|---|---|---|---|---|
| | | | Number of ratios | Minimum ICER (2017 US$ per DALY or QALY) | Maximum ICER (2017 US$ per DALY or QALY) |
| Federated States Of Micronesia | 3867 (716 to 12,300) | 547 | 1 | 16,578 | 16,578 |
| Fiji | 3343 (620 to 10,620) | 815 | 1 | 570 | 570 |
| Guam | 7725 (1415 to 24,619) | 260 | 0 | NA | NA |
| Indonesia | 4796 (884 to 15,347) | 324 | 20 | 25 | 23,563 |
| Kiribati | 2477 (462 to 7746) | 1688 | 5 | 350 | 1973 |
| Laos | 439 (81 to 1401) | 371 | 12 | 216 | 2074 |
| Malaysia | 5534 (1016 to 17,700) | 250 | 2 | cost-saving | 2767 |
| Maldives | 7224 (1322 to 22,868) | 123 | 1 | 3195 | 3195 |
| Marshall Islands | 3404 (632 to 10,786) | 762 | 0 | NA | NA |
| Mauritius | 5671 (1041 to 18,115) | 233 | 7 | cost-saving | 4439 |
| Myanmar | 380 (70 to 1208) | 534 | 9 | 46 | 1196 |
| North Korea | 445 (82 to 1422) | 367 | 9 | 125 | 1122 |
| Northern Mariana Islands | 6114 (1125 to 19,442) | 430 | 0 | NA | NA |
| Papua New Guinea | 315 (58 to 992) | 858 | 6 | 23 | 432 |
| Philippines | 4986 (919 to 15,918) | 287 | 1 | 1746 | 1746 |
| Samoa | 4493 (829 to 14,399) | 380 | 1 | 4216 | 4216 |
| Seychelles | 4837 (895 to 15,280) | 744 | 6 | cost-saving | 5425 |
| Solomon Islands | 347 (64 to 1097) | 674 | 6 | 33 | 382 |
| Sri Lanka | 6840 (1255 to 21,621) | 130 | 9 | 99 | 1998 |
| Taiwan (Province Of China) | 8727 (1597 to 27,873) | 181 | 7 | 1975 | 41,631 |
| Thailand | 4969 (915 to 15.902) | 396 | 33 | 62 | 40,110 |
| Timor-Leste | 4798 (885 to 15,339) | 317 | 9 | 173 | 1887 |
| Tonga | 3795 (702 to 12,105) | 589 | 1 | 3469 | 3469 |
| Vanuatu | 3582 (664 to 11,364) | 666 | 1 | 1865 | 1865 |
| Vietnam | 4866 (897 to 15,543) | 303 | 87 | cost-saving | 21,134 |
| Sub-Saharan Africa | | | | | |
| Angola | 3593 (666 to 11,433) | 675 | 12 | cost-saving | 1973 |
| Benin | 386 (71 to 1222) | 537 | 12 | cost-saving | 1480 |
| Botswana | 4051 (748 to 12,928) | 529 | 7 | cost-saving | 3329 |
| Burkina Faso | 355 (65 to 1117) | 675 | 12 | cost-saving | 1973 |
| Burundi | 379 (69 to 1183) | 688 | 12 | cost-saving | 1233 |
| Cameroon | 386 (71 to 1225) | 519 | 12 | cost-saving | 2589 |
| Cape Verde | 4629 (854 to 14,819) | 349 | 7 | cost-saving | 1726 |
| Central African Republic | 303 (56 to 938) | 1118 | 12 | 123 | 3946 |
| Chad | 360 (66 to 1136) | 638 | 12 | 123 | 3452 |
| Comoros | 313 (58 to 983) | 896 | 12 | cost-saving | 986 |
| Congo (Brazzaville) | 296 (55 to 929) | 1002 | 12 | cost-saving | 636 |
| Cote D'Ivoire | 504 (92 to 1604) | 262 | 12 | cost-saving | 2343 |
| Djibouti | 334 (62 to 1054) | 743 | 12 | cost-saving | 4439 |
| Dr Congo | 347 (64 to 1086) | 759 | 12 | 81 | 1860 |
| Equatorial Guinea | 4409 (811 to 14,029) | 459 | 7 | cost-saving | 11308 |
| Eritrea | 305 (56 to 951) | 1014 | 12 | cost-saving | 4562 |

*(Continued)*

**Table 2.** (Continued)

| Country | Predicted ICER adjusted for cost-saving probabilities (2017 US$ per DALY Averted) | Cervical cancer DALYs per 100 000 women ages 15+ years | Tufts registry dataset plus sensitivity analyses extracted | | |
|---|---|---|---|---|---|
| | | | Number of ratios | Minimum ICER (2017 US$ per DALY or QALY) | Maximum ICER (2017 US$ per DALY or QALY) |
| Ethiopia | 432 (79 to 1371) | 420 | 12 | 30 | 3576 |
| Gabon | 4099 (756 to 13,061) | 526 | 7 | cost-saving | 2096 |
| Ghana | 380 (70 to 1205) | 536 | 12 | cost-saving | 1356 |
| Guinea | 314 (58 to 982) | 927 | 12 | cost-saving | 2084 |
| Guinea-Bissau | 341 (63 to 1071) | 751 | 12 | cost-saving | 1850 |
| Kenya | 448 (82 to 1431) | 360 | 12 | cost-saving | 2836 |
| Lesotho | 307 (57 to 966) | 918 | 12 | cost-saving | 2836 |
| Liberia | 396 (73 to 1249) | 532 | 12 | cost-saving | 1480 |
| Madagascar | 342 (63 to 1069) | 779 | 12 | cost-saving | 2096 |
| Malawi | 371 (68 to 1167) | 627 | 12 | cost-saving | 1233 |
| Mali | 430 (79 to 1369) | 406 | 12 | cost-saving | 1480 |
| Mauritania | 384 (71 to 1219) | 524 | 12 | cost-saving | 1480 |
| Mozambique | 340 (63 to 1066) | 770 | 12 | cost-saving | 1184 |
| Namibia | 4350 (803 to 13,980) | 426 | 7 | cost-saving | 5055 |
| Niger | 379 (70 to 1190) | 612 | 12 | 123 | 3822 |
| Nigeria | 421 (77 to 1345) | 412 | 20 | cost-saving | 17764 |
| Rwanda | 372 (68 to 1177) | 589 | 12 | cost-saving | 1603 |
| Sao Tome and Principe | 342 (63 to 1079) | 708 | 12 | cost-saving | 2219 |
| Senegal | 365 (67 to 1155) | 604 | 12 | cost-saving | 1603 |
| Sierra Leone | 375 (69 to 1179) | 614 | 12 | cost-saving | 1726 |
| Somalia | 344 (63 to 1085) | 1028 | 12 | 62 | 3329 |
| South Africa | 3787 (700 to 12,059) | 622 | 8 | cost-saving | 7326 |
| South Sudan | 321 (59 to 1009) | 830 | 0 | NA | NA |
| Swaziland | 3606 (668 to 11,471) | 666 | 7 | cost-saving | 1973 |
| Tanzania | 370 (68 to 1171) | 592 | 12 | cost-saving | 1233 |
| The Gambia | 437 (80 to 1383) | 414 | 12 | cost-saving | 1480 |
| Togo | 382 (70 to 1205) | 569 | 12 | cost-saving | 1973 |
| Uganda | 414 (76 to 1312) | 460 | 12 | cost-saving | 1356 |
| Zambia | 350 (65 to 1107) | 660 | 12 | cost-saving | 1603 |
| Zimbabwe | 331 (61 to 1043) | 784 | 12 | cost-saving | 1110 |

Predictions for each country were based on GDP per capita, cervical cancer DALYs per capita, and vaccine cost. All country predictions used vaccine coverage of 70% (median across all studies), a bivalent vaccine, target sex of females only, health sector payor perspective, 3% discount rate for costs and health outcomes, lifetime time horizon, DALYs averted as the health outcome measure, null comparator, and less than 100% access to cervical cancer treatment. ICER = incremental cost-effectiveness ratio; DALY = disability-adjusted-life-year; GDP = gross domestic product per capita in 2017 US$.

is higher than the mean vaccine cost of the final sample. Also, the predictions assume a health care payer perspective rather than societal. [**S4 Appendix**].

Globally, the adjusted mean predicted ICER is 2017 US$4217 per DALY averted (95% UI): US$773–13,448). The lowest adjusted mean ICERs are in Congo (Brazzaville) (2017 US$296 per DALY averted; 95% UI: $55–929), Central African Republic ($303 per DALY averted; 95% UI: $56–938), Eritrea ($305 per DALY averted; 95% UI: $56–951), and Lesotho ($307 per DALY averted; 95% UI: $57–966). (**Table 2**) These four countries are in the top 4% of cervical cancer burden rates globally. The highest adjusted mean ICERs are in the USA (2017 US $27,600 per DALY averted; 95% UI: $5,041–88,339), Kuwait (US$17,504 per DALY averted;

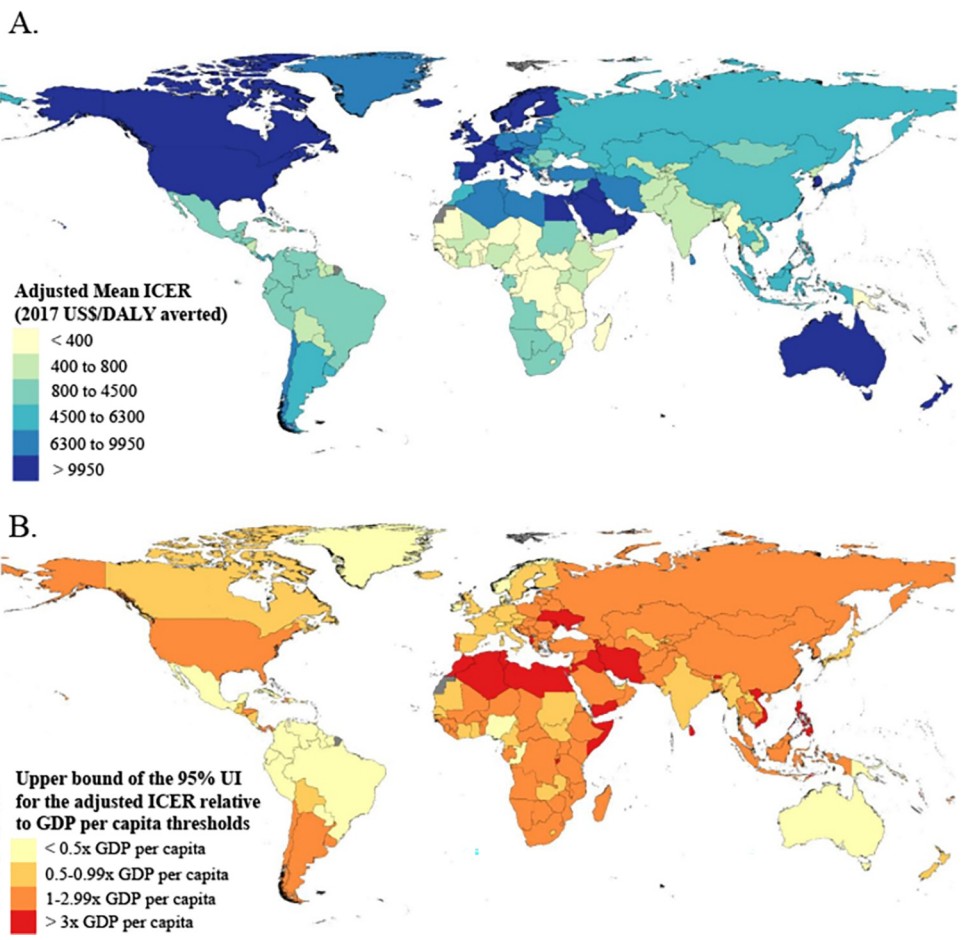

**Fig 3.** (A) Predicted ICERs from the meta-regression analysis by country in 2017 US$ per DALY averted, and (B) predicted ICERs relative to four categories of GDP per capita: <0.5, 0.5 to 0.9, 1.0 to 3.0, and >3.0 times GDP per capita. Results for (B) incorporate the ICER uncertainty intervals such that the upper bound of the uncertainty interval (97.5th percentile) must be within the categories. UI = uncertainty interval; ICER = incremental cost-effectiveness ratio; DALY = disability-adjusted-life-year; GDP = gross domestic product per capita in 2017 US$.

95% UI: $3,169–55,442), Qatar (US$16,070 per DALY averted; 95% UI: $2,907–51,447), and Saudi Arabia (US$15,911 per DALY averted; 95% UI: $2,888–49,937). These four countries, all are in the bottom 25% of cervical cancer burden rates globally.

The adjusted mean ICER is less than 2017 US$400 per DALY averted for 35 countries and it is between 2017 US$400 and $800 for another 29 countries. Among these 64 countries, 38 (59%) are in Sub-Saharan Africa super-region, and 12 (19%) are in Latin America and Caribbean (**Fig 3A**). Fifty-five of 64 (85%) are Gavi-eligible. Among the 38 countries with a mean ICER greater than $9,950 per DALY averted, 27 (71%) are in the High income super-region, and 10 (26%) are in North Africa and Middle East.

Viewing the results in the context of each country's economy, and accounting for uncertainty, the upper bound of the 95% UI for the adjusted ICER is below one-half times GDP per capita for 28 countries (**Fig 3B**). Eighteen of 28 (64%) countries are in the Latin America and Caribbean super-region, and seven (25%) are in High income. Three of 28 countries are Gavi-eligible: Congo (Brazaville), Papau New Guinea, and Nigeria. The upper bound is below one times GDP per capita for an additional 52 countries, including 18 (35%) in the High income super-region, 12 (23%) in Sub-Saharan Africa, 10 (19%) in Latin America and the Caribbean,

**Table 3. Predicted incremental cost-effectiveness ratios aggregated to super-region level and compared to range of input data from Tufts registry dataset and additional extractions.**

| Super-region | Predicted ICER adjusted for cost-saving probabilities (2017 US$ per DALY Averted) | Tufts registry dataset plus sensitivity analyses extracted | | | |
|---|---|---|---|---|---|
| | | Minimum ICER (2017 US$ per DALY or QALY) | Minimum ICER location | Maximum ICER (2017 US$ per DALY or QALY) | Maximum ICER location in Tufts data |
| Central Europe, Eastern Europe, and Central Asia | 5,023 (923 to 16,095) | 25 | Ukraine | 50,565 | Hungary |
| High Income | 14,667 (2,677 to 46,917) | cost-saving | Argentina, Chile, Germany, Norway, Spain | 331,568 | Iceland |
| Latin America and Caribbean | 1,031 (189 to 3,280) | cost-saving | Barbados, Brazil, Guyana, Nicaragua, Venezuela | 77,007 | Colombia |
| North Africa and Middle East | 6,928 (1,266 to 21,841) | cost-saving | Algeria, Bahrain, Kuwait, Morocco, Oman, Qatar, United Arab Emirates | 617,462 | Qatar |
| South Asia | 489 (90 to 1,557) | cost-saving | India, Pakistan | 6,238 | Bangladesh |
| Southeast Asia, East Asia, and Oceania | 5,097 (937 to 16,281) | cost-saving | Mauritius, Seychelles, Vietnam | 78,478 | China |
| Sub-Saharan Africa | 706 (130 to 2,245) | cost-saving | Angola, Benin, Botswana, Burkina Faso, Burundi, Cameroon, Cape Verde, Comoros, Congo, Cote D'Ivoire, Djibouti, Equatorial Guinea, Eritrea, Gabon, Ghana, Guinea, Guinea-Bissau, Kenya, Lesotho, Liberia, Madagascar, Malawi, Mali, Mauritania, Mauritius, Mozambique, Nambia, Nigeria, Rwanda, Sao Tome and Principe, Senegal, Seychelles, Sierra Leone, South Africa, Swaziland, Tanzania, The Gambia, Togo, Uganda, Zambia, Zimbabwe | 13,560 | Nigeria |

Super-region predictions are the population-weighted average of the adjusted mean ICER for the countries in it. Predictions for each country were based on GDP per capita, cervical cancer DALYs per capita, and vaccine cost. All country predictions used vaccine coverage of 70% (median across all studies), a bivalent vaccine, target sex of females only, health sector payor perspective, 3% discount rate for costs and health outcomes, lifetime time horizon, DALYs averted as the health outcome measure, null comparator, and less than 100% access to cervical cancer treatment. ICER = incremental cost-effectiveness ratio; DALY = disability-adjusted-life-year; GDP = gross domestic product per capita in 2017 US$.

and six (12%) in Southeast Asia, East Asia, and Oceania. Eighteen of 52 countries are Gavi-eligible. The upper bound is above three times GDP per capita for 26 countries, including 10 (38%) in the North Africa and Middle East super-region, and eight (31%) in Southeast Asia, East Asia, and Oceania. Three of 26 countries are Gavi-eligible: Burundi, Somalia, and Yemen.

At the GBD super-region level, adjusted mean ICERs are lowest for South Asia and Sub-Saharan Africa, with a population-weighted, adjusted mean ICERs across five countries of $489 per DALY averted (95% UI: $90–1557) and across 46 countries of US$706 per DALY averted (95% UI: $130–2,245) (Table 3), respectively. Adjusted mean ICERs are highest in High income, and North Africa and Middle East, with a population-weighted, adjusted mean ICERs across 34 countries of US$14,667 per DALY averted (95% UI: US$2,677–46,917), and across 21 countries of US$6,928 per DALY averted; 95% UI: $1,266–21,841), respectively.

## Discussion

To our knowledge, this is the first meta-regression analysis of published CEAs, which uses the HPV vaccine as an example for transferring CEA results across settings. We built on published CEA in the Tufts registries, then extracted and exploited their one-way sensitivity analyses to estimate the effects of four covariates on the ICER. The final model estimates included GDP

per capita, and burden of disease at the country-level, four intervention-level covariates, and six methods-level covariates. Vaccine cost is subject to change with policy decisions in the public and private sector, and the meta-regression estimates support straightforward predictions with alternative vaccine costs by location.

Meta-regression analyses are well-known for clinical evidence synthesis, and less well-known for economics research. Decision-makers have reasons to distrust results from a single study, and be concerned about the replicability of published research. Ioannides has argued that false positive findings are more likely to occur in research with specific characteristics [29], and empirical economics research has many of these characteristics [30]. Neumann et al. identified at least one of these characteristics in the CEA literature on pharmaceutical interventions; findings were more likely to be favorable when research was sponsored by a pharmaceutical or device manufacturer [31]. Meta-regression analyses of CEA may ultimately enhance the credibility of CEA research, as well as support transferring results across settings.

Our findings show that the adjusted mean ICER for HPV vaccination is 2017 US$4,217 per DALY averted (95% UI: US$773–13,448) globally, and below US$800 per DALY averted for 64 countries. Our results provide evidence for introducing and expanding HPV vaccination, albeit with substantial uncertainty for some countries. To meet the vaccine target for the WHO Strategy for Cervical Cancer Elimination, progress is needed in incorporating HPV vaccines into national vaccination schedules. Gavi, the Vaccine Alliance subsidizes the vaccine cost to eligible countries, but many of them have not introduced HPV vaccination. The adjusted mean ICER is less than US$800 per DALY averted in 55 of 57 Gavi-eligible countries, but only 30 (55%) are currently receiving HPV vaccine support. Accounting for uncertainty, when the upper bound of the 95% UI is less than one times GDP per capita, we can be reasonably sure that the HPV vaccine is of good value within the context of a country's economy. Eleven (52%) Gavi-eligible countries are not receiving HPV vaccine support among 21 where the upper bound is less than one times GDP per capita: Congo (Brazzaville), Comoros, Djibouti, India, Lesotho, Myanmar, Nicaraqua, Nigeria, Papua New Guinea, South Sudan, and Sudan. Nineteen (42%) are not receiving support among 33 where the upper bound is between one and three times GDP per capita.

Recent HPV vaccine supply chain shortages are a major barrier to increasing vaccine introduction in Gavi-eligible countries. These shortages led to a 65% reduction in Gavi, the Vaccine Alliance's HPV vaccination target of vaccinating 40 million girls by 2020 [32]. Based on forecasts of global demand for HPV vaccines through 2030, current supply under the base case scenario is insufficient to meet demand through 2024 [33]. This is leading countries to postpone the introduction of HPV vaccination, and threatens progress towards achieving the targets outlined in the WHO's cervical cancer elimination initiative. Notable increases in product development and improvements in supply allocation will be critical in ensuring HPV vaccine access and coverage increase in high-burden settings where HPV vaccine introduction is lagging.

Despite there being more published cost-effectiveness estimates for HPV vaccines than any other health intervention, the results of most countries with more than one estimate are heterogeneous. Our meta-regression analysis helps to overcome this challenge by producing a set of standardized ICERs for HPV vaccines in 195 countries after controlling for variation due to each country's epidemiological and economic context, intervention characteristics, and study methods. The between-study heterogeneity drives the uncertainty in our estimates and is due to the lack of standardization across study methods, data sources, and model assumptions. In particular, we found considerable heterogeneity in medical cost-savings and indirect costs (also known as productivity costs). For example, we discovered divergent assumptions about access to care, that has not been addressed in cost-effectiveness recommendations [34]. In

LMICs where little is known about access to screening and treatment for cervical cancer, and consequently the treatment cost saved by preventive interventions, modelers who assume 100% access to treatment for stage four cervical cancer likely over-state the potential savings.

Our analysis is limited in that we were unable to capture all of the method and intervention-level differences between articles in our models. To better understand additional factors in ICER variation, we would need more detailed reporting and data extraction. Specifically, we found that many articles did not report the exact parameters or data sources they were using for access to cancer treatment in their models, making it challenging for us to accurately capture these differences. We also recommend extracting data in the future on whether the models were static or dynamic, and whether or not the models included catch-up campaigns.

Another limitation is that we were unable to include cost-saving ratios in the meta-regression model. The magnitude of the numeric value of cost-saving ratios can not be consistently interpreted. As decision-makers strive to minimize the incremental cost and maximize the incremental health outcomes, the numerator and denominator, respectively, both drive the ICER in opposite directions. As such, we can only derive meaningful relationships between covariates and ICERs with positive incremental costs and positive incremental health outcomes. We treated cost-savings as a binary outcome in our logistic regression model. We did not propagate the correlation structure between this logistic regression model and the main meta-regression. This limitation is unlikely to change our results, because of the low probability that the HPV vaccine is cost-saving. We considered modeling the incremental costs and incremental health outcomes jointly instead of modeling their ratio, and we decided against this approach for two reasons. First, only 274 of 638 (43%) registry entries report numerator and denominator. Second, in an initial comparison of the two approaches with the same sample size, the model fit was worse for the numerator and denominator approach than the ICER approach. If analysts consistently report the incremental costs and incremental health outcomes, and they are extracted into the registries, further comparisons of the approaches are warranted.

We also had to impute the uncertainty for published ICERs. Rather than reporting uncertainty intervals, most CEA studies report the sensitivity of their analyses to various input parameters. Given that ICERs are associated with uncertainty due to measurement error in input parameters, and variation due to methodological choices such as discount rates, time horizon, reporting ICER uncertainty is crucial in allowing these sources of uncertainty to be disentangled in our meta-regression analysis.

Finally, our analysis included ICERs that captured intervention effects on cervical cancer burden alone, as this was the most common health outcome in the published CEA on HPV vaccines. The vaccines have beneficial effects on a wide range of health outcomes, such as anogenital warts, and oropharyngeal, anal, and vicinal cancers. Predicting ICERs that capture all of these health outcomes would provide a more complete picture of the costs and health outcomes associated with HPV vaccines.

## Conclusions

This is the first attempt to generate a complete and consistent set of ICERs for HPV vaccines with UI for 195 countries. Meta-regression analysis can be conducted on CEA, where the one-way sensitivity analyses are used to quantify the effects of factors at the intervention and method-level. There is substantial uncertainty in the predicted ICERs in some countries, due to underlying heterogeneity of published CEA. Our results however, identified countries where the HPV vaccine is a good value, despite the uncertainty, and can facilitate decision-making across a wide range of settings.

Globally, introducing the HPV vaccine and achieving high HPV vaccine coverage are critical steps to eliminating cervical cancer burden. Building on all available information, our results support introducing and expanding HPV vaccination, especially in many countries that are eligible for subsidized vaccines from Gavi, the Vaccine Alliance, and the Pan American Health Organization. Vaccine cost is a key covariate, and our estimated models can be readily predictions ICERs and UI whenever vaccine subsidies are extended to additional countries or the vaccine price changes.

## Supporting information

**S1 Table. GATHER compliance checklist.**
(DOCX)

**S2 Table. Selected characteristics of cost-effectiveness articles on human papillomavirus vaccines included in the analysis.**
(DOCX)

**S1 Appendix. Intervention taxonomy.**
(DOCX)

**S2 Appendix. Data extractions and mapping.**
(DOCX)

**S3 Appendix. Meta-regression analysis appendix.**
(PDF)

**S4 Appendix. Cost-saving predictions.**
(DOCX)

**S5 Appendix. Vaccine cost for predictions.**
(DOCX)

## Author Contributions

**Conceptualization:** Marcia R. Weaver, Christopher J. L. Murray.

**Data curation:** Katherine L. Rosettie, Jonah N. Joffe, Gianna W. Sparks, Shirley Chen, Samuel B. Ewald, Paola Pedroza Velandia, Lauryn Stafford, Marcia R. Weaver.

**Formal analysis:** Katherine L. Rosettie, Jonah N. Joffe, Gianna W. Sparks, Edwin B. Mathew.

**Funding acquisition:** Christopher J. L. Murray.

**Investigation:** Katherine L. Rosettie, Christopher J. L. Murray.

**Methodology:** Katherine L. Rosettie, Jonah N. Joffe, Aleksandr Aravkin, Paola Pedroza Velandia, Peng Zheng, Marcia R. Weaver, Christopher J. L. Murray.

**Project administration:** Kelly Compton.

**Resources:** Christopher J. L. Murray.

**Software:** Aleksandr Aravkin, Peng Zheng.

**Supervision:** Marcia R. Weaver.

**Validation:** Katherine L. Rosettie.

**Visualization:** Katherine L. Rosettie, Gianna W. Sparks, Danielle Michael, Paola Pedroza Velandia.

**Writing – original draft:** Katherine L. Rosettie, Jonah N. Joffe.

**Writing – review & editing:** Jonah N. Joffe, Aleksandr Aravkin, Kelly Compton, Danielle Michael, Molly B. Miller-Petrie, Peng Zheng, Marcia R. Weaver, Christopher J. L. Murray.

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
