## [Decision Letter · Decision Letter 0]

10 May 2021

PONE-D-20-32128

Cost-effectiveness of HPV vaccination in 195 countries: A meta-regression analysis

PLOS ONE

Dear Dr. Christopher JL Murray,

Thank you for submitting your manuscript to PLOS ONE. After careful consideration, we feel that it has merit but does not fully meet PLOS ONE’s publication criteria as it currently stands. Therefore, we invite you to submit a revised version of the manuscript that addresses the points raised during the review process.

Dear author, the urgency of this type of study makes us efficient and quick in our decisions, so I consider that the improvements are minimal as well as the reviewers, so we encourage you to complete your improvements to proceed to its publication.

We look forward to receiving your revised manuscript.

Kind regards,

Carlos Alberto Zúniga-González, Ph.D

Academic Editor

PLOS ONE

Additional Editor Comments:

The minor revisions are in style, I consider their results interesting, so I suggest in the regression metadata analysis approach consider the following references that could help to strengthen your manuscript. a) Dios-Palomares, R. (2015). 7. Analysis of the Efficiency of Farming Systems in Latin America and the Caribbean Considering Environmental Issues. Revista Cientifica-Facultad de Ciencias Veterinarias, 25(1). b) Blanco-Orozco, N., Arce-Díaz, E., & Zúñiga-Gonzáles, C. (2015). Integral assessment (financial, economic, social, environmental and productivity) of using bagasse and fossil fuels in power generation in Nicaragua. Revista Tecnología en Marcha, 28(4), 94-107. c) Zuniga González, C. (2020). Total factor productivity growth in agriculture: Malmquist index analysis of 14 countries, 1979-2008. Revista Electrónica De Investigación En Ciencias Económicas, 8(16), 68-97. https://doi.org/10.5377/reice.v8i16.10661

Journal Requirements:

4. We note that Figure 3 in your submission contain map images which may be copyrighted. All PLOS content is published under the Creative Commons Attribution License (CC BY 4.0), which means that the manuscript, images, and Supporting Information files will be freely available online, and any third party is permitted to access, download, copy, distribute, and use these materials in any way, even commercially, with proper attribution. For these reasons, we cannot publish previously copyrighted maps or satellite images created using proprietary data, such as Google software (Google Maps, Street View, and Earth). For more information, see our copyright guidelines: http://journals.plos.org/plosone/s/licenses-and-copyright.

You may seek permission from the original copyright holder of Figure 3 to publish the content specifically under the CC BY 4.0 license. 

If you are unable to obtain permission from the original copyright holder to publish these figures under the CC BY 4.0 license or if the copyright holder’s requirements are incompatible with the CC BY 4.0 license, please either i) remove the figure or ii) supply a replacement figure that complies with the CC BY 4.0 license. Please check copyright information on all replacement figures and update the figure caption with source information. If applicable, please specify in the figure caption text when a figure is similar but not identical to the original image and is therefore for illustrative purposes only.

Reviewers' comments:

Reviewer's Responses to Questions

**Comments to the Author**

1. Is the manuscript technically sound, and do the data support the conclusions?

Reviewer #1: Partly

Reviewer #2: Yes

2. Has the statistical analysis been performed appropriately and rigorously? 

Reviewer #1: Yes

Reviewer #2: I Don't Know

3. Have the authors made all data underlying the findings in their manuscript fully available?

Reviewer #1: Yes

Reviewer #2: Yes

4. Is the manuscript presented in an intelligible fashion and written in standard English?

Reviewer #1: Yes

Reviewer #2: Yes

5. Review Comments to the Author

Reviewer #1: The author must add a sub section in the methods parts which is about Meta Regression analysis.

Otherwise, the metholodogical part of your paper become weak. Also, the conclusion part must be improved

Reviewer #2: The manuscript seems interesting . The authors have conducted a thorough literature review, and analysed information accurately and sufficiently. However, the manuscript needs revisions.

1.Different formats have been adopted to quote references, the style should be according to the PLOS requirements and uniform.

2.The policy implications need attention. There is an ample room to suggest more policy implications in the conclusion section.

6. PLOS authors have the option to publish the peer review history of their article (what does this mean?). If published, this will include your full peer review and any attached files.

Reviewer #1: **Yes: **DURSUN BALKAN

Reviewer #2: No

---

## [Author Response · Author response to Decision Letter 0]

1 Nov 2021

3.1. The author must add a sub section in the methods parts which is about Meta Regression analysis. Otherwise, the metholodogical part of your paper become weak. 

Response: In the revised subsection on Modelling approaches, we expanded the description of the five stages of the mixed-effects meta-regression framework from one paragraph to six, with a paragraph devoted to each of the five stages, as reproduced below: 

“The statistical model and fitting procedures for the analysis of ICERs was conducted in five stages, and used a mixed-effects meta-regression framework (MR-BRT).20 This model included priors on all covariates and a study-specific random intercept. Each stage is described briefly below; for further information, see S3 Appendix.

“In the first stage, we estimated priors for selected covariates by leveraging the fact that one-way sensitivity analyses differ in no unmeasured covariates from their reference analyses. Four covariates had a sufficient number of sensitivity analyses reported published CEA to estimate priors using crosswalk models: vaccine cost, vaccine coverage, cost discount rate, and discount rate for health outcomes. We matched each sensitivity analysis with its corresponding reference analysis, and the crosswalk model estimated the difference in log-ICERs between sensitivity and reference analyses as a function of the difference between values of that covariate. We then constructed Gaussian priors for these covariates to use in all subsequent stages of the analysis with means and standard deviations equal to the crosswalk parameter estimates and standard errors from these crosswalk models. 

“In the second stage, we estimated a nonlinear response curve for log-GDP per capita by modeling the log-ICERs as a nonlinear function of log-GDP per capita. Log cervical cancer DALYs per capita was entered linearly into this model, in addition to the four covariates addressed in the first stage, and the priors calculated in the first stage were placed on the corresponding covariates. To make this stage more robust to model misspecification, we placed a spline ensemble on log GDP per capita. This model also used a robust statistical approach for outlier detection, and outliers trimmed at this stage were discarded from subsequent steps of the analysis. The nonlinear response curve estimated by this model was used to transform log-GDP per capita for use in subsequent stages of the analysis.

“In the third stage, we selected additional covariates to include in the final meta-regression using a generalized Lasso approach for linear mixed effects models. The four crosswalk covariates, log cervical cancer DALYs per capita, and spline-transformed log-GDP per capita were pre-selected covariates at this stage, and the priors estimated for the crosswalk covariates were placed on those covariates. This process selected from nine additional candidate covariates: target sex, the proportion of model population assumed to have access to cervical cancer treatment, vaccine type, perspective, time horizon, comparator, and the outcome measure, and whether or not the intervention included a booster dose. Only one of these covariates, the assumption of a booster dose, was not selected for inclusion in the final model.

“In the fourth stage we selected the standard deviation of a Gaussian prior to apply to all regression parameters other than the intercept and the parameters for the four crosswalk covariates. To select a standard deviation, we fit a mixed effects meta-regression models with random intercepts by study, and priors on crosswalk covariates as calculated in the first stage. We normalized all other covariates and included Gaussian priors on those covariates, centered at zero and with a standard deviation that was constant across covariates. We varied this standard deviation using a grid-search and used 10-fold cross-validation to select the standard deviation that minimized the mean squared error for predicting values in the holdout set. We then converted the prior standard deviation back to the unstandardized scale for each covariate.

“In the fifth stage, we fit a mixed effects model with a random intercept and priors on covariates determined in the first and fourth stages. This model included priors on covariates calculated in the first and fourth stages and the transformed version of log-GDP per capita, and random intercepts by study.” 

3.2 Also, the conclusion part must be improved

Response: The conclusion has been revised as follows: 

This is the first attempt to generate a complete and consistent set of ICERs for HPV vaccines with UI for 195 countries. Meta-regression analysis can be conducted on CEA, where the one-way sensitivity analyses are used to quantify the effects of factors at the intervention and method-level. There is substantial uncertainty in the predicted ICERs in some countries, due to underlying heterogeneity of published CEA. Our results however, identified countries where the HPV vaccine is a good value, despite the uncertainty, and can facilitate decision-making across a wide range of settings.

Globally, introducing the HPV vaccine and achieving high HPV vaccine coverage are critical steps to eliminating cervical cancer burden. Building on all available information, our results support introducing and expanding HPV vaccination, especially in many countries that are eligible for subsidized vaccines from Gavi, the Vaccine Alliance, and the Pan American Health Organization. Vaccine cost is a key covariate, and our estimated models can be readily predictions ICERs and UI whenever vaccine subsidies are extended to additional countries or the vaccine price changes. 

Reviewer #2: The manuscript seems interesting . The authors have conducted a thorough literature review, and analysed information accurately and sufficiently. However, the manuscript needs revisions.

4.1. Different formats have been adopted to quote references, the style should be according to the PLOS requirements and uniform.

Response: Thank you for this helpful feedback. The format for citing references complies with the PLOS One requirements in the revised manuscript. 

4. 2. The policy implications need attention. There is an ample room to suggest more policy implications in the conclusion section.

Response: Please see the response to comment 3.2 above.

---

## [Decision Letter · Decision Letter 1]

18 Nov 2021

Cost-effectiveness of HPV vaccination in 195 countries: A meta-regression analysis

PONE-D-20-32128R1

Dear Dr. Christopher JL Murray,

We’re pleased to inform you that your manuscript has been judged scientifically suitable for publication and will be formally accepted for publication once it meets all outstanding technical requirements.

Kind regards,

Carlos Alberto Zúniga-González, Ph.D

Academic Editor

PLOS ONE

Additional Editor Comments (optional):

Congratulations dear authors for the effort to improve the quality of your manuscript.

Reviewers' comments:

Reviewer's Responses to Questions

**Comments to the Author**

1. If the authors have adequately addressed your comments raised in a previous round of review and you feel that this manuscript is now acceptable for publication, you may indicate that here to bypass the “Comments to the Author” section, enter your conflict of interest statement in the “Confidential to Editor” section, and submit your "Accept" recommendation.

Reviewer #2: All comments have been addressed

2. Is the manuscript technically sound, and do the data support the conclusions?

Reviewer #2: Yes

3. Has the statistical analysis been performed appropriately and rigorously? 

Reviewer #2: Yes

4. Have the authors made all data underlying the findings in their manuscript fully available?

Reviewer #2: Yes

5. Is the manuscript presented in an intelligible fashion and written in standard English?

Reviewer #2: Yes

6. Review Comments to the Author

Reviewer #2: The topic of the study appears to be interesting. The review comments have been addressed adequately. I wish the author(s) all the best.

7. PLOS authors have the option to publish the peer review history of their article (what does this mean?). If published, this will include your full peer review and any attached files.

Reviewer #2: No

---

## [Editor Report · Acceptance letter]

10 Dec 2021

PONE-D-20-32128R1 

Cost-effectiveness of HPV vaccination in 195 countries: A meta-regression analysis 

Dear Dr. Murray:

I'm pleased to inform you that your manuscript has been deemed suitable for publication in PLOS ONE. Congratulations! Your manuscript is now with our production department. 

Kind regards, 

on behalf of

Dr. Prof. Carlos Alberto Zúniga-González 

Academic Editor

PLOS ONE